# Transcriptomic Analysis Reveals Sixteen Potential Genes Associated with the Successful Differentiation of Antibody-Secreting Cells through the Utilization of Unfolded Protein Response Mechanisms in Robust Responders to the Influenza Vaccine

**DOI:** 10.3390/vaccines12020136

**Published:** 2024-01-29

**Authors:** Ahmed Tawfik, Takahisa Kawaguchi, Meiko Takahashi, Kazuya Setoh, Izumi Yamaguchi, Yasuharu Tabara, Kristel Van Steen, Anavaj Sakuntabhai, Fumihiko Matsuda

**Affiliations:** 1Functional Genetics of Infectious Diseases Unit, Institut Pasteur, CNRS UMR2000, 75015 Paris, France; moustafa.ahmed.6r@kyoto-u.jp; 2Pasteur International Unit at Center for Genomic Medicine, Kyoto University, Kyoto 606-8507, Japan; 3Center for Genomic Medicine, Graduate School of Medicine, Kyoto University, Kyoto 606-8507, Japanyamaguchi@genome.med.kyoto-u.ac.jp (I.Y.);; 4BIO3—Laboratory for Systems Genetics, GIGA-R Medical Genomics, University of Liège, 4000 Liège, Belgium; 5BIO3—Laboratory for Systems Genetics, GIGA-R Medical Genomics, University of Leuven, 3000 Leuven, Belgium; 6Ecology and Emergence of Arthropod-Borne Pathogens Unit, Institut Pasteur, CNRS UMR2000, 75015 Paris, France

**Keywords:** influenza vaccines, influenza vaccine response, humoral immune response, B cell response, transcriptomic analysis, causal genes

## Abstract

The seasonal influenza vaccine remains one of the vital recommended infection control measures for the elderly with chronic illnesses. We investigated the immunogenicity of a single dose of influenza vaccine in 123 seronegative participants and classified them into four distinct groups, determined by the promptness of vaccine response, the longevity of humoral immunity, and the likelihood of exhibiting cross-reactivity. Subsequently, we used transcriptional profiling and differential gene expression analysis to identify potential genes directly associated with the robust response to the vaccine. The group of exemplary vaccine responders differentially expressed 16 genes, namely: MZB1, MYDGF, TXNDC5, TXNDC11, HSP90B1, FKBP11, PDIA5, PRDX4, CD38, SDC1, TNFRSF17, TNFRSF13B, PAX5, POU2AF1, IRF4, and XBP1. Our findings point out a list of expressed proteins that are related to B cell proliferation, unfolded protein response, and cellular haemostasis, as well as a linkage of these expressions to the survival of long-lived plasma cells.

## 1. Introduction

The influenza virus is a transmissible respiratory viral pathogen that can have widespread implications on the global population during its seasonal outbreak. According to updated estimations from the World Health Organization (WHO), influenza viruses are responsible for an annual mortality rate that ranges from 290,000 to 650,000 due to the development of severe respiratory diseases. Individuals of all age groups, irrespective of their overall health, can experience complications caused by influenza [1]. Furthermore, individuals belonging to specific high-risk categories, including the elderly and those with chronic conditions, are at an increased risk of developing more severe complications such as fulminant pneumonia [2]. Influenza vaccines are associated with decreased rates of laboratory-confirmed influenza illness, as well as clinical influenza-like illness, among children [3,4,5], healthy adults under the age of 65 [6,7,8], and healthy older adults [9]. Nevertheless, influenza vaccines are prone to specific limitations, such as the adverse impact on vaccine effectiveness caused by inter-individual variability of humoral immune responses, the production of antibodies with limited cross-reactivity against non-selected strains in the annual vaccine composition, and lack of long-lasting protection due to influenza virus antigenic drifts [10]. Therefore, it is imperative to conduct further research studies to acquire a deeper understanding of the immune response mechanisms that yield robust responses to influenza vaccines.

In Japan, a study was conducted in 2011/2012 to assess the immunogenicity of a single dose of the trivalent inactivated influenza vaccine (TIIV) (A/H1N1, A/H3N2, and B) in healthy Japanese adults. Hemagglutination-inhibiting titre measurements were obtained at three time points: before vaccination and 7 and 90 days post-vaccination, and transcriptomics measurements were obtained at two time points: before vaccination and 7 days post-vaccination. The aim of this study was to unveil transcriptional signatures and potential genes associated with the host immune mechanisms responsible for inducing a promising response to influenza vaccines. We assessed seroprotection using serum strain-specific antibodies and identified a group of robust vaccine responders who demonstrated an ideal response characterized by promptness, cross-reactivity, and longevity. Consequently, by undertaking a comparative analysis of the transcriptional signatures of the group of ideal responders alongside three other groups that demonstrated partial, weak, and non-existent vaccine responses, we ascertained a list of 16 potential differential genes related to the ideal immune response; these included two extracellular cytokines, MZB1 and MYDGF, six cytoplasm enzymes, TXNDC5, TXNDC11, HSP90B1, FKBP11, PDIA5, and PRDX4, two plasma membrane enzymes, CD38 and SDC1, two transmembrane receptors, TNFRSF17 and TNFRSF13B, and four known transcription regulators, PAX5, POU2AF1, IRF4, and XBP1. The functional enrichment and protein–protein interaction network of these expressed genes confirmed that robust vaccine responders possess transcriptional signatures that are significantly associated with B cell proliferation and differentiation, as well as metabolic processes related to the stress response to protein folding and the maintenance of antibody secretion.

## 2. Materials and Methods

### 2.1. Study Design

A research study was conducted during the 2011/2012 influenza season to assess the immunogenicity of a single dose of TIIV (A/California/7/2009 (H1N1) pdm09, A/Victoria/210/2009 (H3N2), and B/Brisbane/60/2008) and to identify the transcriptional signature associated with the ideal response to the vaccine in healthy adults. The vaccine compositions included the virus strains that were circulating in Japan and overseas during the influenza season of 2010/2011 and were anticipated for the season of 2011/2012. Influenza viruses isolated during the actual influenza season of 2011/2012 were in accordance with the vaccine compositions and predominantly comprised subtype H3N2 (71%) and type B (28%). Influenza type B viruses, comprising Victoria and Yamagata lineages, were isolated in a 2:1 ratio. There were a few incidences of H1N1pdm09 (0.2%).

The study participants included 123 Japanese healthy individuals residing in Nagahama City, Japan: 78 (63.41%) female and 45 (36.56%) male participants, aged between 37 and 66 years (Mean = 58.11 and Standard Deviation = 7.0). The participants received a TIIV single dose (1 mL/vial) on the first day and were then followed up for three months. One blood sample was taken prior to vaccination at day 0 for the purpose of transcriptomic profiling and baseline immunogenicity assessment. The baseline immunogenicity assessment confirmed that all participants were seronegative for the three virus strains included in the vaccine. Two blood samples were taken at day 7 and day 90 post vaccination, for two main objectives. The primary objective was to assess transcriptomic profiling and short-term immunogenicity one week after vaccination, while the secondary objective was to evaluate long-term immunogenicity three months after vaccination and its correlation with a transcriptomic signature measured at day 7.

Prior to vaccination, the participants wrote an informed written consent and answered a questionnaire about their overall health, allergic status, and previous doses of influenza vaccinations. The study protocol was approved by the Ethics Committee of Kyoto University Graduate School and Faculty of Medicine.

### 2.2. Haemagglutination Inhibition Assay

The influenza virus is an enveloped virus with a segmented single-strand RNA genome of negative polarity. Trivalent inactivated vaccines encompass the process of virus purification, resulting in the retention of viral haemagglutinin (HA) and neuraminidase (NA) glycoproteins [11] derived from two subtypes of influenza A viruses (H1N1 and H3N2) and one subtype of influenza B viruses. HA is a homotrimeric glycoprotein responsible for the attachment of the virus to the surface of the host cell by binding it to sialic acid receptors [12]. The vaccine administration generates haemagglutination-inhibiting antibodies as an immune response that prevents infection by disrupting the binding of the virus to host receptors [13]. A hemagglutination-inhibiting (HAI) antibodies immunological assay (SRL, Inc., Tokyo, Japan), also called a HAI titre, is the most common serological test available for assessing the influenza vaccine response. A HAI titre value of ≥1:40 is generally accepted to be associated with a 50% reduction in the risk of illness [14] and could be regarded as a seroprotection threshold value. 

### 2.3. Transcriptomic Profiling and Differential Gene Expression Analysis

RNA was extracted and hybridized using Agilent one colour microarray technology. Following quality control and data normalization of raw measurements, differential gene expression analysis was performed using the Limma (Linear Models for MicroArray data) R package [15]. Limma applies gene-wise linear models to microarray data and simultaneously estimates logarithmic ratios between different transcriptomic samples. The differential analysis was conducted between the samples taken at day 7 and day 0. For a result to be considered as statistically significant and associated with the vaccine response, an adjusted *p*-value cut-off of 0.05 was applied using the Benjamin and Hochberg (BH) procedure, along with an absolute log2-fold change cut-off value of 0.6, indicating a change in expression exceeding 1.5 times. We used the analysis of variance model [16], which is included within the stat R package, to determine the significant variations in gene expressions’ log2-fold change among the four subgroups exhibiting different response patterns to the vaccine. Tukey’s honest significant difference test generated confidence intervals for the differences between the mean log2-fold changes among the subgroup. Finally, we utilized gene ontology (GO) enrichment and protein–protein interaction (PPI) networks to interpret the differentially expressed transcripts. Specifically, we utilized the clusterProfiler R package [17] with default configurations to enrich GO molecular processes. Additionally, we utilized the STRING proteins database [18] to construct the PPI network, which was retrieved, visualized, and enriched using the Cytoscape visualization tool with default settings.

## 3. Results

### 3.1. Vaccination Immunogenicity

Initially (day 0), a total of 123 (100%) seronegative participants received a single dose of TIIV. One week post-vaccination (day 7), we tested for the presence of HAI antibody titres and detected vaccine-induced seroprotection against at least one virus strain in 86 (69.91%) participants. Specifically, 54 (43.9%) acquired immunity against the H1N1 strain, 47 (38.21%) against the H3N2 strain, and 49 (39.83%) against the B1 strain. Three months post-vaccination (day 90), vaccine-induced seroprotection was detected in 113 participants (91.87%). Specifically, 68 (55.28%) acquired immunity against the H1N1 strain, 76 (61.78%) against the H3N2 strain, and 91 (73.98%) against the B1 strain. The variation in immune responses has resulted in the identification of two groups that showed inadequate responses. Firstly, it was observed that 72 (58.53%) participants did not acquire seroprotection against one or more virus strains, both at day 7 and day 90. These individuals could be described as non-responders to at least one of the vaccine strains. Secondly, 21 (17.04%) participants experienced a loss of seroprotection against one or more virus strains, which they had acquired at day 7 before undergoing the second test at day 90. These individuals could be described as short-term responders to at least one of the vaccine strains. However, there were also two groups that experienced adequate responses. Firstly, 76 participants (61.78%), who were not seroprotected against one or more virus strains at day 7, were able to seroconvert at some point between day 7 and day 90. These individuals could be described as late responders to at least one vaccine strain. Secondly, 81 participants (65.85%), who had acquired seroprotection against one or more virus strains at day 7, maintained the acquired seroprotection until day 90. These individuals could be described as ideal responders to at least one of the vaccine strains (Table 1). The ideal response, in this case, is characterized by both promptness and longevity, as seroprotection was acquired within a short span of 7 days and persisted for a minimum of 90 days. Our focus was primarily on the ideal responders rather than the late responders; we aimed to identify the transcriptomic signature associated with the ideal responses occurring within one week, as the transcriptomic profiling was feasible at day 7.

To conduct a comprehensive analysis of the overall vaccine response, we categorized participants according to the number of virus strains effectively brought under control by the vaccine-induced humoral immunity (Figure 1). A small group of 11 participants (8.94%) was identified as having responded ideally to the vaccine by achieving seroprotection against all three virus strains at Day 7 and maintaining this seroprotection until Day 90. A partially protected group of 24 (19.51%) participants demonstrated ideal vaccine responses to two virus strains. Specifically, ten (8.13%) participants achieved seroprotection against H1N1 and H3N2, eight (6.5%) participants achieved seroprotection against H1N1 and B1, and six (4.88%) participants achieved seroprotection against H3N2 and B1. A weakly protected group of 46 (37.98%) participants demonstrated ideal vaccine responses to one virus strain only: 16 (13.0%) against H1N1, 13 (10.57%) against H3N2, and 17 (13.82%) against B1. Finally, an unprotected group of 42 (34.15%) participants who have demonstrated an inability to achieve the ideal response against any strain of the virus, suggesting that a majority of participants within this group remained vulnerable to infection, despite receiving vaccination.

### 3.2. Transcriptional Signatures of Fully Protected Participants

The vaccine immunogenicity assessment showed that the group of fully protected individuals was the smallest. Our main objective was to understand the transcriptomic profiles after vaccination of this particular group and analyse the transcriptional signatures that played a role in their rapid immune response and durable seroprotection. Transcriptomic profiling was performed twice, prior to vaccination at day 0 and post-vaccination at day 7. Through differential expression analysis, we identified 1849 differentially expressed transcripts between day 7 and day 0 in the group of fully protected participants, of which 708 transcripts were up-regulated and 1141 transcripts were down-regulated. The compilation of 1849 differentially expressed transcripts comprised a wide array of immunoglobulin genes, alongside a varied range of transcript types that included cytokines, transmembrane receptors, transcription regulators, and enzymes. In contrast, 1430, 857, and 784 transcripts were differentially expressed in the groups of partial, weak, and no protection, respectively. Only 316 transcripts were overlapped between the four groups and 1057 transcripts were expressed only in the group of fully protected participants (Appendix A).

An unsupervised hierarchical cluster analysis was performed on 1849 transcripts that were differentially expressed in the fully protected group and revealed two distinct clusters, which reflected significant changes in gene expression levels prior to vaccination at day 0 and post-vaccination at day 7 (Figure 2A). This was further supported by a principal component analysis on the same set of transcripts, comparing all the participants, and revealed a distinct cluster that separated the fully protected group from the other groups (Figure 2B).

GO enrichment of molecular processes (Appendix A) provided a straightforward interpretation of the differently expressed transcripts in the fully protected group and highlighted the main biological processes and molecular pathways involved in the influenza vaccine response mechanism (Figure 3). The process of B cell activation was enriched (adjusted *p*-value = 1.7 × 10^−8^); this process is pivotal to the vaccine response and involves sophisticated pathways and signalling cascades upon binding to antigens derived from the inactivated influenza virus strains [19,20,21]. The following several biological processes were enriched, indicating the activation of B cells due to the recognition of antigens and the transmission of activation signals through the B-cell receptor (BCR): the B cell receptor signalling pathway (adjusted *p*-value = 9.3 × 10^−12^), phagocytosis recognition (adjusted *p*-value = 1.2 × 10^−11^), the antigen receptor-mediated signalling pathway (adjusted *p*-value = 3.3 × 10^−7^), and the immune response-activating cell surface receptor signalling pathway (adjusted *p*-value = 1.5 × 10^−5^). These processes emphasize the need to promptly react to an antigenic stimulus, as well as the availability of a diverse repertoire of naïve B cells in order to achieve a successful activation of B cells. The significantly differentially expressed transcripts that contributed to the enrichment of these processes included FCRL1 and CD40. The expression of FCRL1 attains its highest levels in naive B cells, and then decreases once B cells become activated and start forming germinal centres [22]. On the other hand, the expression of CD40 indicates a possible interaction with helper T cells that influence naive B cell development and activation via CD40 signalling [23].

The BCR dependent signal, stimulated by an antigen, also triggers the development of B lymphoblasts, which divide and differentiate into short-lived plasmablasts that secrete antibodies [24]. This initial extrafollicular response produces early protective antibodies of slightly lower affinity towards the inactivated influenza virus strains and with shorter life span. Furthermore, B cells possess the indispensable ability to phagocyte and process antigens to present antigenic peptides on B cell surfaces, via major histocompatibility complex (MHC) class II molecules. This serves as a communication facility between activated B cells and follicular helper T cells (T_FH_), which have the same antigen-specificity and reside near the B cell follicles [25,26]. Once T_FH_ cells come into contact with B cells that display identical linked antigen epitopes on their MHC class II molecules, T_FH_ cells start to help by expressing their CD40 ligand and secreting various cytokines that promote the survival and proliferation of B cells. Moreover, under the influence and support of T_FH_ cells, germinal centres (GCs) begin to form and activated follicular B cells begin to undergo affinity maturation to effectively target specific viral epitopes [27,28,29,30,31,32]. The germinal centres form several days after an antigen encounter as temporary structures in the centre of the B cell follicles and are characterized by intense cellular activity involving cell division and cell death. This region is surrounded by a network of follicular dendritic cells (FDCs) [33], a group of activated and resting B cells, and antigen-specific T_FH_ cells. The germinal centre itself comprises two primary areas: the light zone and the dark zone. Within the dark zone, activated B cells undergo proliferation and somatic hypermutation (SHM) to adapt their specific immunoglobulin variable regions, which make up the antigen-binding sites of their B cell receptor, in order to increase affinity and better align with epitopes present in influenza virus proteins, which include the surface hemagglutinin (HA) glycoprotein, the primary target of the vaccine. 

B cells with higher affinity BCRs move to the light zone and attempt to capture antigens trapped on FDCs for further processing and presenting on their MHC class II molecules. Alternatively, B cells with lower affinity BCRs are unable to effectively capture and present antigens. Consequently, these B cells ultimately undergo apoptosis and die due to lack of support from T_FH_. On the other hand, successfully proliferating B cells undergo repeated cycles of entry to the dark zone for continuous somatic hypermutations and affinity selection. The germinal centre formation and reactions enriched the biological process, termed adaptive immune response, based on the somatic recombination of immune receptors built from immunoglobulin superfamily domains (adjusted *p*-value = 1.7 × 10^−4^). The significantly differentially expressed transcripts that contributed to the enrichment of this process included TLR4, IL4, and BCL6. The expression of TLR4 modulates FDC activation and maturation and plays a role in preventing germinal centre B cell apoptosis [34]. On the other hand, the expression of BCL6 is regulated by IL4 cytokine, while BCL6 acts as a critical mediator of the magnitude and duration of the germinal centre responses [35].

Ultimately, B cells exit the germinal centre, either as long-lived plasma cells (LLPCs) or memory B cells (MBCs) [36]. The former, LLPCs, migrate to the bone marrow and compete for dedicated survival niches that enable them to sustain antibody secretion for a prolonged duration [37,38,39,40,41,42]. Meanwhile, MBCs retain their B cell characteristics in anticipation of encountering the same antigen and are programmed to rapidly differentiate into antibody-secreting cells (ASCs) of high affinity [43,44]. The generation of these two B cell lineages and the production of antibodies are the expected outcomes of successful vaccination, which led to the enrichment of two significant biological processes that indicate the production of antibodies through biosynthesis after exposure to a stimulus; a humoral immune response mediated by circulating immunoglobulin (adjusted *p*-value = 4.7 × 10^−14^) and immunoglobulin production (adjusted *p*-value = 6.9 × 10^−9^). The significantly differentially expressed transcripts that contributed to the enrichment of these processes included MZB1 and XBP1. Both transcripts code for multifunctional proteins that are involved in the differentiation of B cells into plasma cells and the proper adaptation and functioning of antibody-secreting cells. These two transcripts are among the list of potential genes that were found to contribute to the ideal vaccine response, with further explanation and details provided in the following section.

### 3.3. Genes with Significant Variations in log2-Fold Change between the Fully Protected Group and the Non-Fully Protected Groups

We analysed the transcriptional signature of the fully protected group and compared each significant transcript’s log2-fold change with those of the three groups that did not achieve full protection—namely, the partially protected, weakly protected, and non-protected groups. Through an analysis of variance (Appendix A), we identified a list of 16 genes that showed significant pair-wise differences in their log2-fold change means between the fully protected group and the other three groups, indicating their potential influence on the effectiveness of the vaccine response (Figure 4). The identified 16 genes were CD38, SDC1, MZB1, TNFRSF17, TNFRSF13B, POU2AF1, PAX5, IRF4, TXNDC5, TXNDC11, HSP90B1, FKBP11, PDIA5, MYDGF, PRDX4, and XBP1.

Considering the significant variations in the log2-fold change observed among the groups, we attempted to investigate the known functions of each gene to gain insight and comprehend the potential impact of these genes on the ideal vaccine response.

CD38 was up-regulated within the fully protected group, with an average log2-fold change of 2.95. In contrast, the partially, weakly, and non-protected groups demonstrated statically significant reductions, with average log2-fold changes of 1.64, 1.36, and 1.15, respectively. Moreover, the significant differences in the average log2-fold changes, when comparing the fully protected group to the partially, weakly, and non-protected groups, as indicated by the calculated confidence intervals, were 1.31 [0.33, 2.27] (adjusted *p*-value = 1.5 × 10^−3^), 1.59 [0.68, 2.47] (adjusted *p*-value = 1.1 × 10^−5^), and 1.8 (0.89, 2.69) (adjusted *p*-value = 1.1 × 10^−6^), respectively. CD38, also referred to as CADPR1, is a multifunctional protein expressed on the surface of B cells in healthy individuals. It acts as both a receptor and a multifunctional enzyme, and one of its key roles is to catalyse the synthesis and hydrolysis of a general calcium messenger molecule cyclic ADP-ribose (cADPR) [45]. Recent research has revealed that CD38 has a strong connection with CD19 in inactive B cells, as well as with the immunoglobulin M (IgM) B cell receptor when it is in an engaged state, suggesting a modulatory effect on B cell activation upon antigen recognition by regulating CD19 [46]. Furthermore, researchers used CD19, CD38, and CD138 to classify different plasma cell subsets in the bone marrow of human subjects, and their findings revealed that the CD19^−^CD38^++^CD138^+^ subset was morphologically distinct and exclusively contained plasma cells that targeted viral antigens, which the subjects had not encountered for over four decades as a result of durable immunisation [47]; consequently, elevated levels of CD38 expression may indicate the effective production of LLPCs.SDC1 was up-regulated within the fully protected group, with an average log2-fold change of 1.46. In contrast, the partially, weakly, and non-protected groups demonstrated statically significant reductions, with average log2-fold changes of 0.79, 0.73, and 0.48, respectively. The significant differences in the average log2-fold changes, when comparing the fully protected group and the non-fully protected groups, were 0.67 [0.12, 1.2] (adjusted *p*-value = 3.9 × 10^−3^), 0.73 [0.23, 1.22] (adjusted *p*-value = 4.6 × 10^−4^), and 0.98 [0.47, 1.47] (adjusted *p*-value = 1.6 × 10^−6^), respectively. SDC1, also referred to as the CD138 antigen, encodes a heparan sulphate glycoprotein. SDC1 is a member of the syndecan proteoglycan family, which mediates cell signalling, cell binding, and cell migration. CD138 serves as a cell surface marker for normal B cells and is expressed at varying levels throughout different stages of B cell differentiation [48]. A higher expression of CD138 indicates the presence of LLPCs. Notably, the expression of CD138 on ASCs leads to increased levels of heparan sulphate, known for its ability to bind pro-survival cytokines like IL-6 and APRIL, to protect ASCs from apoptosis, and to promote longevity [49].MZB1 was up-regulated and exhibited a remarkable increase in expression level within the fully protected group, with an average log2-fold change of 3.66. In contrast, the partially, weakly, and non-protected groups demonstrated statically significant reductions, with average log2-fold changes of 2.36, 2.04, and 1.7, respectively. The significant differences in the average log2-fold changes, when comparing the fully protected group and the non-fully protected groups, were 1.3 [0.24, 2.37] (adjusted *p*-value = 3.4 × 10^−3^), 1.62 [0.64, 2.61] (adjusted *p*-value = 3.6 × 10^−5^), and 1.96 [0.97, 2.95] (adjusted *p*-value = 1.4 × 10^−6^), respectively. MZB1, also referred to as pERp1, encodes the marginal zone B and the B1 cell specific protein. It is found within the endoplasmic reticulum (ER) as a component of the binding immunoglobulin protein (BiP) chaperone complex. MZB1 belongs to the canopy (CNPY) family of ER resident saposin-like proteins, and it has a saposin fold, with unique sequence extensions that are not present in other saposin proteins [50]. During the process of the B cell to plasma cell differentiation, MZB1 is significantly up-regulated, and analyses demonstrated that MZB1 is an unusual type of resident ER protein that specifically assists immunoglobulin biosynthesis. Knowing this, plasma cells have the ability to produce and release huge amounts of immunoglobulin molecules that undergo assembly and oxidative folding within the ER [51]. Other studies revealed that the deletion of MZB1 adversely affects humoral immune responses and the secretion of antibodies in plasma cells that naturally undergo ER stress [52]. In addition, the retention of ASCs in the bone marrow and their maturation into plasma cells requires the involvement of a cell surface molecule known as very late antigen 4 (VLA4). Interestingly, ASCs deficient in the co-chaperone MZB1, which is essential for VLA4 activation, showed an impaired ability to migrate and home in the bone marrow [53].TNFRSF17 was up-regulated within the fully protected group, with an average log2-fold change of 3.09. In contrast, the partially, weakly, and non-protected groups demonstrated statically significant reductions, with average log2-fold changes of 1.63, 1.3, and 1.06, respectively. The significant differences in the average log2-fold changes, when comparing the fully protected group and the non-fully protected groups, were 1.46 [0.25, 2.67] (adjusted *p*-value = 4.6 × 10^−3^), 1.79 [0.67, 2.9] (adjusted *p*-value = 8.2 × 10^−5^), and 2.03 [0.9, 3.16] (adjusted *p*-value = 3.1 × 10^−5^), respectively. TNFRSF17, also referred to as BCMA or TNFRSF13A, encodes the tumour necrosis factor (TNF) receptor superfamily member 17 protein. It is a non-glycosylated integral membrane protein that is preferentially expressed in mature B lymphocytes. TNFRSF17 interacts with TNF receptor-associated factors TRAF1, TRAF2, and TRAF3, leading to the activation of nuclear factor kappa B (NF-κB), elk-1, c-Jun N-terminal kinase, and p38 mitogen-activated protein kinase [54]. B-cell activating antigen (BCMA), transmembrane activator and CAML interactor (TACI), and B-cell activating factor receptor (BAFFR) serve as receptors for the B-cell activating factor (BAFF) and a proliferation-inducing ligand (APRIL), establishing a complex network that plays a crucial role in the progression of humoral immunity. More specifically, the functional activity of the BCMA receptor aids in promoting the survival of LLPCs [55,56,57,58]. The investigation into the genetic knockout of TNFRSF17 yielded noteworthy findings, specifically a substantial reduction in ASCs in the bone marrow when compared to wild-type controls, occurring 6–8 weeks after immunisation. Nevertheless, germinal centre responses and early antigen-specific serum IgM and IgG levels remained within normal ranges, indicating that the primary impact of losing BCMA affected the LLPCs [58].TNFRSF13B was up-regulated within the fully protected group, with an average log2-fold change of 2.22. In contrast, the partially, weakly, and non-protected groups demonstrated statically significant reductions, with average log2-fold changes of 1.17, 1.07, and 1.05, respectively. The significant differences in the average log2-fold changes, when comparing the fully protected group and the non-fully protected groups, were 1.05 [0.04, 2.06] (adjusted *p*-value = 0.2 × 10^−2^), 1.15 [0.22, 2.1] (adjusted *p*-value = 4.2 × 10^−3^), and 1.17 [0.23, 2.13] (adjusted *p*-value = 4.1 × 10^−3^), respectively. TNFRSF13B, also referred to as TACI, encodes a lymphocyte-specific member of the TNF receptor superfamily. The TNF superfamily ligands BAFF and APRIL, along with their three receptors BAFFR, BCMA, and TACI, play significant roles in the immunological functions of the B cell arm of the immune system. BAFF-R specifically targets BAFF, while BCMA has a greater affinity for APRIL than BAFF. On the other hand, TACI is capable of binding both ligands with equal effectiveness. TACI facilitates NF-κB responses and triggers the process of immunoglobulin IgG and IgA class-switch recombination in B cells. In humans, TACI deficiency was found to manifest as an antibody deficiency syndrome [59,60,61,62,63].POU2AF1 was up-regulated within the fully protected group, with an average log2-fold change of 0.88. In contrast, the partially, weakly, and non-protected groups demonstrated statically significant reductions, with average log2-fold changes of 0.36, 0.18, and 0.05, respectively. The significant differences in the average log2-fold changes, when comparing the fully protected group and the non-fully protected groups, were 0.52 [0.05, 1.0] (adjusted *p*-value = 0.1 × 10^−2^), 0.7 [0.26, 1.14] (adjusted *p*-value = 8.4 × 10^−5^), and 0.83 [0.39, 1.28] (adjusted *p*-value = 3.2 × 10^−6^), respectively. POU2AF1, also referred to as B cell-specific coactivator (OCA-B) or OBF-1, encodes an octamer-binding factor protein. OCA-B polypeptide is the primary factor that activates immunoglobulin (Ig) promoters in B cells, working in conjunction with OCT-1 and OCT-2 binding proteins; it forms a binding complex that specifically targets octamer sites found in both promoters and enhancers. Its main role is to initiate the transcription process for Ig genes in B cells [64,65,66]. Furthermore, the transcriptional activator OCT-2, along with its cofactor OBF-1, serves as key regulatory factors for IL6 expression, driving the differentiation of activated CD4+ T cells into T_FH_ cells. Therefore, OBF1 plays a role in the response of B cells to thymus-dependent antigens, being essential for the formation of germinal centres that are fundamental to the generation of high-affinity antibody-secreting cells [67,68,69].PAX5 was up-regulated within the fully protected group, with an average log2-fold change of 0.95. In contrast, it was down-regulated within the partially, weakly, and non-protected groups, with average log2-fold changes of −0.2, −0.05, and −0.25, respectively. The significant differences in the average log2-fold changes, when comparing the fully protected group and the non-fully protected groups, were 1.15 [0.0, 2.31] (adjusted *p*-value = 2.3 × 10^−2^), 1.0 [−0.06, 2.07] (adjusted *p*-value = 3.5 × 10^−2^), and 1.2 [0.13, 2.28] (adjusted *p*-value = 9.8 × 10^−3^), respectively. PAX5, also referred to as B cell lineage-specific activator protein (BSAP), encodes a member of the paired box (PAX) family of transcription factors. PAX5 is a unique transcription factor that safeguards the B-lymphocyte lineage commitment and performs a dual role by activating B cell-specific genes and simultaneously repressing B cell-unspecific genes [70]. The pro-B cell stage signifies a B cell lineage commitment phase, wherein the rearrangement of heavy-chain genes takes place. Approximately 23% of all expression alterations observed during the transition from common lymphoid progenitors to committed pro-B cells can be attributed to PAX5-regulated genes, which identifies PAX5 as an essential regulator of the B cell developmental transition [71]. The targets of PAX5 activation include immune receptors, such as CD19 and CD21, as well as transcription factors’ interferon regulators, such as IRF4, IRF8, and BACH2 [72]. The PAX5 repressed genes control a wide range of biological functions, including cell communication, adhesion, migration, nuclear processes, and cellular metabolism as part of the process of B cell commitment [73]. For instance, PAX5 repression of the cohesin release factor WAPL in pro-B cells results in alterations to the chromosomal architecture, which facilitates the generation of a diverse antibody repertoire [74]. Studies also revealed that PAX5 plays a role in proliferation and immunoglobulin isotype switching in germinal centre B cells [75].IRF4 was up-regulated within the fully protected group, with an average log2-fold change of 0.8. In contrast, the partially, weakly, and non-protected groups demonstrated statically significant reductions, with average log2-fold changes of 0.26, 0.14, and 0.12, respectively. The significant differences in the average log2-fold changes, when comparing the fully protected group and the non-fully protected groups, were 0.54 [0.0, 1.07] (adjusted *p*-value = 2.5 × 10^−2^), 0.65 [0.16, 1.15] (adjusted *p*-value = 1.8 × 10^−3^), and 0.67 [0.16, 1.18] (adjusted *p*-value = 9.9 × 10^−4^), respectively. IRF4 encodes a transcription factor belonging to the IRF (interferon regulatory factor) family of transcription factors, characterized by a specific DNA-binding domain and the ability to bind to regulatory elements in promoters of interferon-inducible genes [76]. IRF4 is required for the generation of germinal centre B cells by inducing the expression of key germinal centre genes, including BCL6 and AICDA. IRF4 also induces BLIMP1, which handles the transition from a germinal centre B cell gene expression program to that of a plasma cell. This multifunctional nature of IRF4 implies its involvement in a multifaceted regulatory network, wherein its expression levels play an additional role; lower IRF4 expression levels appear to facilitate the progression of the germinal centre pathway and higher expression levels promote the differentiation of plasma cells [77]. Investigating the involvement of IRF4 in post-germinal centre B cell development revealed that IRF4 plays an important role in the differentiation of plasma cells and the process of class switch recombination (CSR). The conditional deletion of IRF4 in germinal centre B cells lacked post-germinal centre plasma cells and the inability to differentiate memory B cells into plasma cells, which highlights the significance of IRF4 as a pivotal transcriptional regulator in the development of plasma cells [78].TXNDC5 was up-regulated and exhibited a remarkable increase in expression level within the fully protected group, with an average log2-fold change of 4.79. In contrast, the partially, weakly, and non-protected groups demonstrated statically significant reductions, with average log2-fold changes of 3.11, 2.74, and 2.43, respectively. The significant differences in the average log2-fold changes, when comparing the fully protected group and the non-fully protected groups, were 1.68 [0.31, 3.03] (adjusted *p*-value = 4.6 × 10^−3^), 2.05 [0.79, 3.3] (adjusted *p*-value = 5.1 × 10^−5^), and 2.36 [1.09, 3.61] (adjusted *p*-value = 5.6 × 10^−6^), respectively. TXNDC5, also referred to as PDIA15 or ERp46, encodes the thioredoxin domain-containing protein 5 and a member of the protein disulfide isomerase (PDI) family. It plays a role in the formation and rearrangement of disulfide bonds for proper protein folding. In addition, TXNDC5 functions as a molecular chaperone, regulating the synthesis of abnormal proteins and maintaining cellular homeostasis. TXNDC5 possesses various biological functions, including anti-oxidation, the promotion of angiogenesis, cellular inflammation, and energy metabolism [79,80].TXNDC11 was up-regulated within the fully protected group, with an average log2-fold change of 1.24. In contrast, the partially, weakly, and non-protected groups demonstrated statically significant reductions, with average log2-fold changes of 0.7, 0.51, and 0.4, respectively. The significant differences in the average log2-fold changes, when comparing the fully protected group and the non-fully protected groups, were 0.54 [0.03, 1.06] (adjusted *p*-value = 1.5 × 10^−2^), 0.73 [0.26, 1.21] (adjusted *p*-value = 1.7 × 10^−4^), and 0.84 [0.37, 1.32] (adjusted *p*-value = 1.1 × 10^−5^), respectively. TXNDC11, also referred to as EFP1, encodes the thioredoxin domain-containing protein 11. TXNDC11 is specifically induced by ER stress and is regulated by the IRE1–XBP1 pathway. This pathway is a major component of the unfolded protein response (UPR) signalling pathway, which plays a significant role in maintaining cellular homeostasis [81].HSP90B1 was up-regulated within the fully protected group, with an average log2-fold change of 1.47. In contrast, the partially, weakly, and non-protected groups demonstrated statically significant reductions, with average log2-fold changes of 0.71, 0.67, and 0.44, respectively. The significant differences in the average log2-fold changes, when comparing the fully protected group and the non-fully protected groups, were 0.76 [0.18, 1.33] (adjusted *p*-value = 2.4 × 10^−3^), 0.8 [0.27, 1.33] (adjusted *p*-value = 1.6 × 10^−4^), and 1.03 [0.49, 1.56] (adjusted *p*-value = 1.8 × 10^−6^), respectively. HSP90B1, also referred to as GRP94, encodes heat shock protein 90 beta family member 1. It is an important ER molecular chaperone that plays a role in the UPR pathway and enhances the functioning of B cells by chaperoning Toll-like Receptors (TLRs) and integrins. The chaperone function of the HSP90B1 gene is of great significance in protein physiology, as well as in the processing and transportation of secreted proteins [82].FKBP11 was up-regulated within the fully protected group, with an average log2-fold change of 1.41. In contrast, the partially, weakly, and non-protected groups demonstrated lower average log2-fold changes of 0.74, 0.72, and 0.51, respectively. The significant differences in the average log2-fold changes, when comparing the fully protected group and the non-fully protected groups, were 0.67 [0.13, 1.22] (adjusted *p*-value = 3.5 × 10^−3^), 0.69 [0.19, 1.19] (adjusted *p*-value = 1.1 × 10^−3^), and 0.9 [0.39, 1.41] (adjusted *p*-value = 8.6 × 10^−6^), respectively. FKBP11, also referred to as FKBP19, encodes FK506-binding protein 11. It is a member of the FKBP family of peptidyl-prolyl cis/trans isomerases, which accelerate the folding of proteins during protein synthesis. Recent studies have discovered that the differentiation of B cells into plasma cells is accompanied by the induction of FKBP11 expression, suggesting its potential role as a catalyst for antibody folding in plasma cells. In particular, the upregulation of FKBP11 expression has been observed to correlate with the induction of ER stress as part of the UPR pathway and in a manner that is dependent on the X-box-binding protein 1 (XBP1) [83].PDIA5 was up-regulated within the fully protected group, with an average log2-fold change of 1.2. In contrast, the partially, weakly, and non-protected groups demonstrated statically significant reductions, with average log2-fold changes of 0.55, 0.54, and 0.32, respectively. The significant differences in the average log2-fold changes, when comparing the fully protected group and the non-fully protected groups, were 0.65 [0.06, 1.25] (adjusted *p*-value = 1.1 × 10^−2^), 0.66 [0.12, 1.21] (adjusted *p*-value = 4.7 × 10^−3^), and 0.88 [0.33, 1.44] (adjusted *p*-value = 3.2 × 10^−4^), respectively. PDIA5, also referred to as PDIR, encodes disulfide isomerase A5 protein. It is a member of the disulfide isomerase (PDI) family of ER proteins, which catalyse protein folding and thiol–disulfide interchange reactions. The domain organization of PDIR is atypical and different from other well-known members of the PDI family. PDIR stands out as the sole PDI consisting of one N-terminal non-catalytic domain and three catalytic domains. Upon analysing the crystal structure of the non-catalytic domain of human PDIR, it became apparent that this domain serves as the primary binding site for the major ER chaperone calreticulin [84].MYDGF was up-regulated within the fully protected group, with an average log2-fold change of 1.07. In contrast, the partially, weakly, and non-protected groups demonstrated statically significant reductions, with average log2-fold changes of 0.53, 0.51, and 0.34, respectively. The significant differences in the average log2-fold changes, when comparing the fully protected group and the non-fully protected groups, were 0.54 [0.05, 1.05] (adjusted *p*-value = 1.2 × 10^−2^), 0.56 [0.11, 1.03] (adjusted *p*-value = 3.7 × 10^−3^), and 0.73 [0.27, 1.19] (adjusted *p*-value = 5.1 × 10^−4^), respectively. MYDGF, also referred to as C19orf10 or IL-25, encodes paracrine-acting protein. It is highly conserved throughout evolution and can be found in various cellular compartments, including the ER, Golgi apparatus, and extracellular space [85]. MYDGF is produced by mucosal epithelial cells, and when it is overly expressed, it promotes eosinophilia and triggers the production of T_H2_-type cytokines [86]. Furthermore, it has been observed to activate the conventional pathways of the NF-κB through the phosphorylation of NF-κBp65 in germinal centre B cells [87].PRDX4 was up-regulated within the fully protected group, with an average log2-fold change of 0.95. In contrast, the partially, weakly, and non-protected groups demonstrated lower average log2-fold changes of 0.33, 0.29, and 0.12, respectively. The significant differences in the average log2-fold changes, when comparing the fully protected group and the non-fully protected groups, were 0.62 [0.13, 1.11] (adjusted *p*-value = 3.7 × 10^−3^), 0.66 [0.21, 1.11] (adjusted *p*-value = 8.9 × 10^−4^), and 0.83 [0.38, 1.29] (adjusted *p*-value = 6.2 × 10^−6^), respectively. PRDX4, also referred to as AOE372, encodes an antioxidant enzyme and belongs to the peroxiredoxin family. It is located in the ER and plays a role in safeguarding against oxidative stress by detoxifying cellular peroxides [88]. However, the function of this protein goes beyond eliminating peroxide; it also promotes oxidative protein folding through the oxidation of PDI [89]. Furthermore, PRDX4 defines a reduction–oxidation pathway that specifically regulates the activity of NF-κB by modulating the phosphorylation of IκB-alpha in the cytoplasm [90].XBP1 was up-regulated within the fully protected group, with an average log2-fold change of 0.75. In contrast, the partially, weakly, and non-protected groups demonstrated statically significant reductions, with average log2-fold changes of 0.22, 0.31, and 0.14, respectively. The significant differences in the average log2-fold changes, when comparing the fully protected group and the non-fully protected groups, were 0.53 [0.07, 0.99] (adjusted *p*-value = 7.7× 10^−3^), 0.44 [0.02, 0.87] (adjusted *p*-value = 1.6 × 10^−2^), and 0.61 [0.18, 1.04] (adjusted *p*-value = 7.9 × 10^−4^), respectively. It is a basic-region leucine zipper protein in the CREB/ATF (cyclic AMP response element binding protein/activating transcription factor) family of transcription factors. XBP1 functions as a transcription factor during ER stress by facilitating the UPR pathway triggered by accumulated misfolded proteins [91], and it has an additional function in protecting cells against oxidative stress [92]. Studies have revealed that the specific signals responsible for triggering plasma cell differentiation and the UPR pathway operate in synchronization through XBP1 to promote successful terminal B-cell differentiation [93,94]. The activation of XBP1 is important to maintain the optimal differentiation, functioning, and survival of LLPCs with high secretory activity. However, recent studies revealed that the absence of XBP-1 only leads to a reduction, rather than a total cessation of antibody secretion in plasma cells [95].

After thoroughly investigating the documented functionalities of each gene, we proceeded to investigate further and constructed a PPI network using the list of 16 genes. This network substantially aids in the recognition of proteins that directly interact with one another, as well as the identification of the function of closely interconnected proteins. Interestingly, all of the proteins were found to have interactions, resulting in the creation of two densely interacting sub-networks (Figure 4A). One of these sub-networks consisted of eight genes that functionally enriched the gene ontology biological process termed the regulation of B cell proliferation, a process that regulates the rate of B cell proliferation and has a direct correlation with germinal centre reactions and the frequency of terminal differentiation of B cells into ASCs. The other sub-network consisted of eight genes that functionally enriched two gene ontology biological processes, an endoplasmic reticulum unfolded protein response and cellular homeostasis; these two biological processes are closely linked to the essential adaption of the metabolism of ASCs to accommodate the elevated rate of protein biosynthesis required for antibody secretion. At the molecular level, the unfolded protein response denotes a series of molecular signals that emerge as a stress response due to the presence of unfolded proteins within the endoplasmic reticulum. These signals allow for the synchronization of protein synthesis rates with nutrient and energy stores, consequently facilitating the secretion of the required substantial number of antibodies [96]. The functional enrichment of these sub-networks provided insights into the biological processes that, if compromised, could potentially hinder the ideal response to the vaccine.

The network’s topological analysis also indicates that XBP1, IRF4, and HSP90B1 are the central hub nodes with the highest degrees of interactions, and, together, they form the backbone of the network. They are crucial for maintaining the network’s structure and functioning, which implies that any substantial reduction in the production of these proteins might impact the entire protein interaction network and consequently compromise the biological processes that are essential for an ideal vaccine response.

### 3.4. Genes with Significant Variations in log2-Fold Change between Participants Receiving the Vaccine for the First Time and Those Repeatedly Vaccinated

Initially (day 0), a total of 123 seronegative participants received a single dose of TIIV. A group of 32 participants reported having received influenza vaccines in previous influenza seasons, irrespective of their diminished antibody level at day 0. Consequently, we assessed the immunogenicity of the vaccine in accordance with the participants’ vaccination history, and it was observed that individuals who had received multiple previous vaccinations exhibited diminished rates of vaccine response within the confines of this study (Table 2). This outcome was expected, since their previous vaccinations have not stimulated an effective immune response or may have only provided short-term protection.

The fully protected group, consisting of 11 participants who responded ideally to the vaccine, had not previously received the vaccine. Conversely, the non-protected group, consisting of 42 participants who have demonstrated an inability to achieve the ideal response against any strain of the virus, included 14 out of 32 (43.75%) participants who had previously received an influenza vaccine and 28 out of 91 (30.76%) participants who had not previously received any influenza vaccine. Through an analysis of variance of the gene expressions log2-fold change for the 16 potential genes (Appendix A), it was found that the differences in the mean log2-fold change significantly decreased across all genes when comparing the fully protected group to the repeatedly vaccinated non-protected group (Figure 5). The group of 14 participants, who lacked ideal responses to any of the vaccine strains and showed either a short-term or no response after past vaccinations, exhibited the lowest gene expression levels of the potential genes, suggesting that the influence of these genes on the efficacy of the vaccine response is not coincidental, but rather a significant impact.

## 4. Discussion

Influenza vaccines possess a unique characteristic, as they are seasonal vaccines that may necessitate annual updates in order to safeguard against circulating influenza strains and accommodate viral antigenic drifts. The vaccines that are currently licensed have the ability to reduce the incidence of laboratory-confirmed cases of influenza. However, they have two main limitations: firstly, they are unable to induce a cross-reactive adaptive immune response that can effectively neutralize different strains of influenza viruses apart from those included in the vaccine. Secondly, these vaccines do not provide long-lasting seroprotection and may necessitate the administration of additional booster shots.

Recently, there have been endeavours to address the first limitation by developing universal vaccines that induce the production of cross-reactive antibodies against influenza highly conserved proteins. These proteins include the stalk structure of the surface protein hemagglutinin instead of its highly variable head structure [97], along with the neuraminidase surface protein (NA) [98] and matrix protein (M2) [99]. Nonetheless, the second limitation regarding the durability of seroprotection remains a topic of continuous investigation. This delay in resolving the second limitation can be attributed to the methodologies employed in the development of influenza vaccines, which depend on the inactivation or attenuation of the isolated virus strains to elicit the host immune response, subsequently followed by vaccine effectiveness studies. It is worth mentioning that this approach lacks extensive research on the biological processes underlying the immune response required to acquire the desired long-lasting seroprotection.

The objective of our study was to identify and interpret the transcriptional signatures of vaccine receivers who exhibited an exemplary immune response, generating protective antibodies against all virus strains of the vaccine. Their transcriptional signatures imply potential mechanisms for inducing broadly neutralizing antibodies and discerning the biological processes associated with the desired influenza vaccine response, particularly as their differentially expressed transcripts might imply potential casual factors. Our results suggest that the discrepancy in certain gene expression levels is associated with the inter-individual variation in antibody response. The list of potential genes that exhibited significant variations between participants who responded robustly and those who did not respond robustly included CD38 and CD138; both enzymes serve as LLPC surface markers and signify an augmented population of the antibody-secreting LLPC population. A B cell response of this nature, resulting from stimulation by vaccine antigens, normally entails the transformation of naïve B-cells into LLPCs, which necessitates an alteration in the morphology and gene expression profile of B cells to promote antibody secretion as an effector function. This transformation program entails the expansion of the ER and Golgi apparatus to increase the capacity of the immunoglobulin secretory apparatus. In addition, plasma cells consistently activate the UPR pathway, primarily triggered by XBP-1, downstream of Blimp-1. The UPR functions as a homeostatic intracellular signalling network to detect stress and deal with the accumulation of unfolded proteins within the ER lumen; this response may subsequently trigger further expansion of the secretory apparatus capacity [100]. Consequently, plasma cells employ autophagy, an intracellular lysosome-mediated bulk degradation pathway, to act as a feedback control mechanism that limits ER size and antibody secretion by moderating the expression of the transcription factors BLIMP1 and XBP1 [101]. Our findings indicated that ideal vaccine responders exhibited significant expression levels of XBP1, HSP90B1, and TXNDC5 proteins, which are known to participate in the protein processing in the endoplasmic reticulum pathway. Meanwhile, ideal responders exhibited significant expression levels of XBP1, HSP90B1, PDIA5, and MYDGF proteins, which participate in the UPR pathway. 

The generation of long-lived antibody-secreting plasma cells is not merely the terminal stage of the B-cell lineage differentiation, since these cells are intended to persist indefinitely and maintain a consistent level of antibody production throughout the life of the vaccine recipients. In order to accomplish this objective, plasma cells must migrate from secondary lymphoid organs to the bone marrow and attempt to acquire survival niches. The niche provides a microenvironment conducive to receiving multiple survival signals that promote longevity. The niche hosts multiple cellular lineages that secrete survival cytokines including IL-6 and APRIL [102,103,104]. The chemokine CXCL12 and its receptor CXCR4 are instrumental in facilitating the localization of plasma cells within vital survival niches [105]. Plasma cells also express several cell surface proteins, which establish intricate interactions with the surrounding niche microenvironment to ensure their prolonged survival. These surface proteins include BCMA [58], VLA4 [106], CD138 [49], CD28 [107], and CD93 [108]. Knowing this, BCMA receptor and its ligand APRIL are recognized as the most well-studied pathway for the survival of plasma cells, and our results confirmed their importance, as the ideal vaccine responders stood out by exhibiting notably elevated levels of TNFRSF17 expression, which encodes the BCMA protein.

The entire process, beginning with antigen stimulation and culminating in the production of long-lived plasma cells, could be elucidated as a sequence of transitions explained by a gene regulatory network of transcription factors [109]. This network involves transcription factors that control B cell identity and promote the maintenance of B cell fate, which are expressed in mature B cells and suppressed in plasma cells, including PAX5, POU2AF1, IRF8, BCL6, and BACH2. Furthermore, transcription factors govern the transcriptional program specific to plasma cells, including BLIMP-1, IRF4, and XBP1. Among the ideal vaccine responders, PAX5, PAX5, POU2AF1, IRF4, and XBP-1 were up-regulated, while IRF8, BCL6, BACH2, and BLIMP1 were down-regulated. The overall network state suggests the activation of plasma cells, as the repression of PAX5 is unnecessary for plasma cell development and activation [110]. Furthermore, POU2AF1 is essential for ASC differentiation and the generation of ASCs that can secrete antibodies in adequate quantities [69]. The suppression of BLIMP1 could indicate an autophagy’s feedback mechanism to mitigate excessive unregulated antibody secretion [101].

Depicting the exact expression level of these transcription regulators in B cells as they undergo proliferation and differentiation towards the fate of plasma cells is a challenging task, impeded by the fact that the average gene expression levels observed across all cell populations obscures the distinctive phenotype of intermediate stages prior to plasma cell terminal differentiation. However, most importantly, the bulk transcriptomic signatures may indicate the key transcriptional regulators linked to inter-individual variability in vaccine response. Our findings propose the inclusion of PAX5, POU2AF1, IRF4, and XBP1 as key transcriptional regulators. Notably, IRF4 and XBP1 emerge as hub proteins that exhibit multifunctional properties, and diminishing their expression levels could potentially hinder the efficacy of the humoral immune response to the influenza vaccine. 

This study is not the initial investigation of its kind; a prior study conducted by Nakaya et al. [111] aimed to identify predictive gene signatures associated with B cell responses observed at day 7 and day 28. Among their findings, CD38 and TNFRSF17 were included in the list of key genes. Moreover, it is noteworthy to mention that further investigations in the future could greatly benefit from employing longitudinal single-cell transcriptomic analyses of samples obtained from ideal vaccine responders. This approach could potentially unveil the expression levels within each B cell subpopulation and uncover a more detailed regulatory network. Furthermore, the identification of minor populations of LLPCs within B cell subpopulations, coupled with a comprehensive understanding of the intricate molecular processes governing their differentiation and maintenance, will result in the development of improved vaccines that elicit robust immune responses.

## 5. Conclusions

Transcriptomic analysis reveals a direct association of certain gene expression levels after influenza vaccinations, with the production and maintenance of long-lived antibody-secreting cells, which lead to robust and long-standing immune responses. Individuals who had previously been vaccinated and lost their protective antibodies showed a very weak association with the expression levels of the identified genes. Our discoveries indicate potential genes implicated in immunological biological processes that, if taken into account during vaccine development, could accelerate the advancement towards a universal influenza vaccine.

## Figures and Tables

**Figure 1 vaccines-12-00136-f001:**
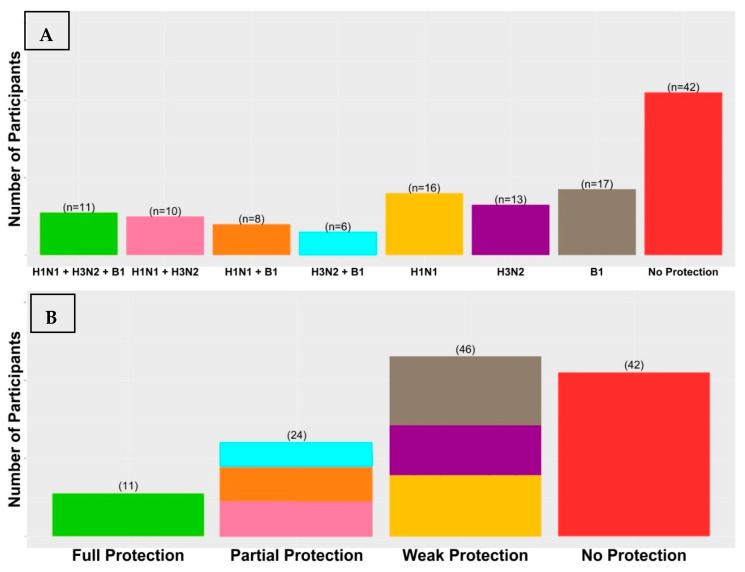
The inter-individual variability in antibody responses after receiving a single dose of TIIV. We classified participants into eight groups according to the development of seroprotection at day 7 which continued to persist for 90 days against the vaccine virus strains (**A**). We further classified participants into four groups according to their level of acquired protection through vaccination and determined by the number of vaccine virus strains they were ideally responsive to (**B**). The findings revealed that a smaller percentage of participants achieved full or partial protection compared to those who had weak or no protection.

**Figure 2 vaccines-12-00136-f002:**
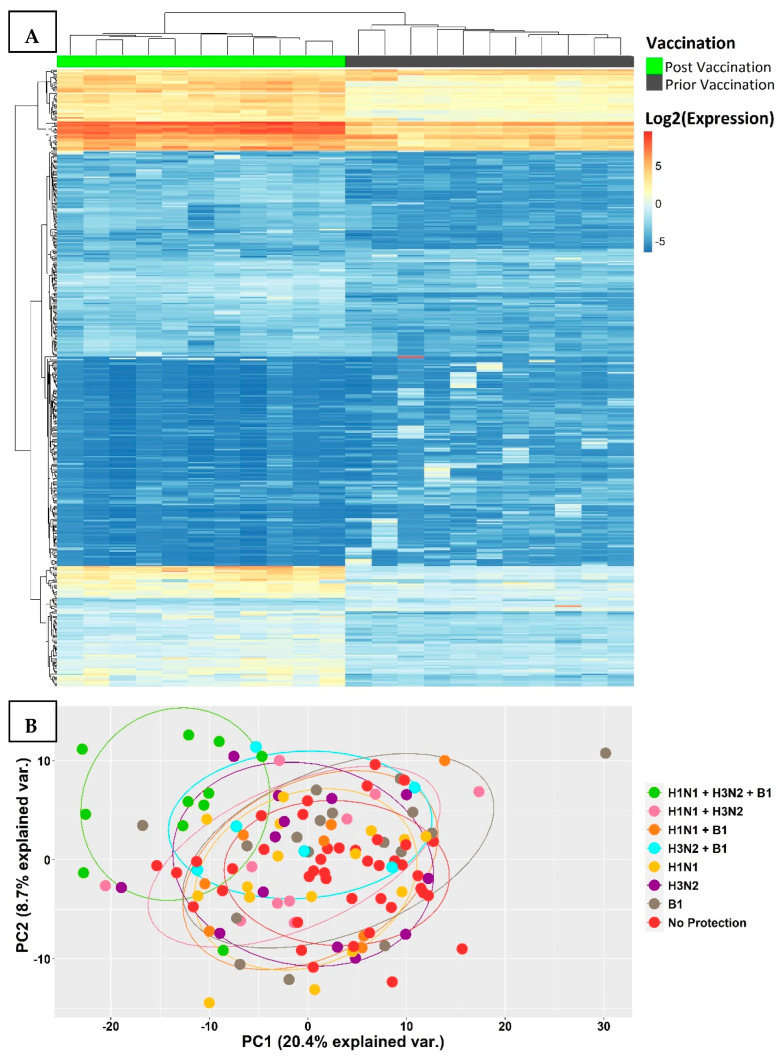
(**A**) A heat map of transcripts differentially expressed between Day 7 and Day 0 showed a remarkable difference in expression levels post-vaccination in all fully protected participants. The transcriptomic samples that were measured prior to vaccination displayed a distinctive cluster towards the right, while the samples collected post-vaccination were clustered towards the left. (**B**) A principal component analysis on differentially expressed transcripts separated the fully protected group in the top left corner from the rest of the groups.

**Figure 3 vaccines-12-00136-f003:**
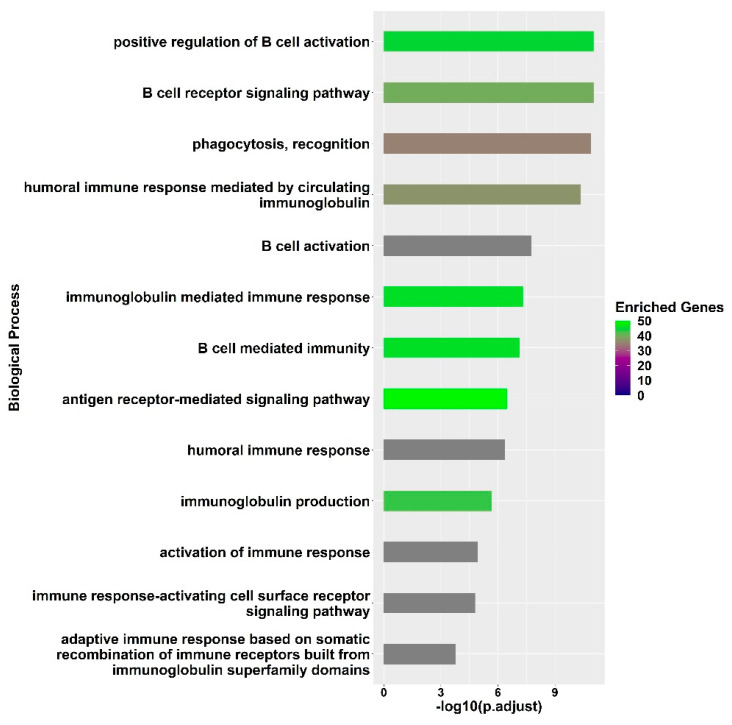
GO enrichment analysis of differentially expressed genes between Day 7 and Day 0 in fully protected participants. The significance of the immunity related processes is represented by the adjusted *p*-value on a log10 scale, while the enrichment genes legend reflects the number of significantly differentially expressed genes found to be associated with the significant biological process.

**Figure 4 vaccines-12-00136-f004:**
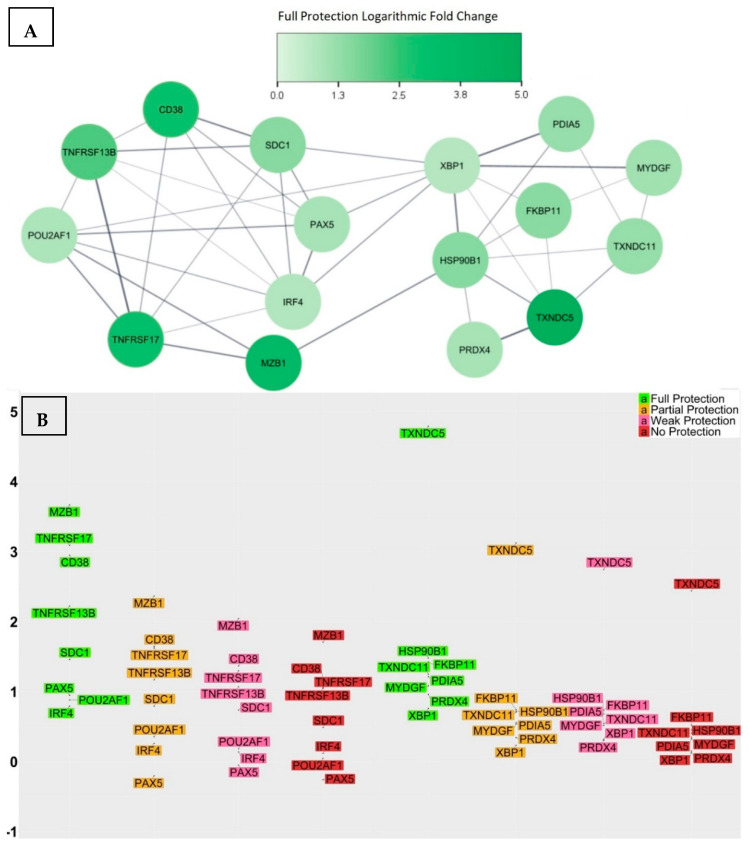
(**A**) The PPI network associated with the 16 genes showed two distinct sub-networks comprising interacting proteins. Among them, XBP1, IRF4, and HSP90B1 emerged as multifunctional hub proteins that bridged the two sub-networks. (**B**) The log2-fold change in gene expression levels at day 7 for all study participants grouped by their level of vaccine-induced protection. The average change in folds appears to decrease in a step-like manner with decreasing levels of protection, with the most significant decrease observed between the fully protected and partially protected groups. This sketch suggests that higher levels of these proteins are crucial for achieving a comprehensive and sustainable vaccine response.

**Figure 5 vaccines-12-00136-f005:**
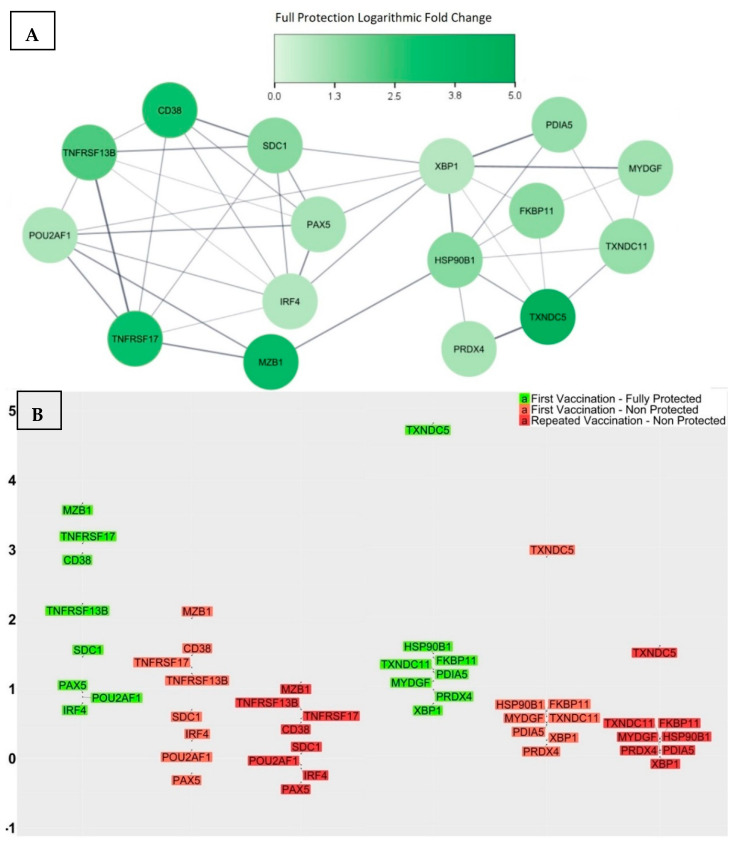
(**A**) The PPI network associated with the 16 genes. (**B**) The log2-fold change in gene expression levels for the groups of fully protected participants (first vaccination), non-protected participants (first vaccination), and non-protected participants (repeated vaccination), respectively. The average change in folds appears to decrease in a step-like manner as the level of protection decreases, as well as in cases where previous vaccinations have been ineffective.

**Table 1 vaccines-12-00136-t001:** Participants’ seroprotection against the vaccine virus strains after 7 and 90 days following the administration of the vaccine.

Seroprotected Participants	H1N1	H3N2	B1
**Day 0**	0 (0.0%)	0 (0.0%)	0 (0.0%)
**Day 7**	54 (43.9%)	47 (38.21%)	49 (39.83%)
**Day 90**	68 (55.28%)	76 (61.78%)	91 (73.98%)
**Neither Day 7 nor Day 90** **(non-responders)**	46 (37.4%)	40 (32.52%)	25 (20.33%)
**Only Day 7** **(short-term responders)**	9 (7.32%)	7 (5.69%)	7 (5.69%)
**Only Day 90** **(late responders)**	23 (18.7%)	36 (29.27%)	49 (39.83%)
**Day 7 and Day 90** **(ideal responders)**	45 (36.58%)	40 (32.52%)	42 (34.15%)

**Table 2 vaccines-12-00136-t002:** Participants’ seroprotection against the vaccine virus strains at 7 and 90 days post-vaccination. The data presented are categorized according to participants’ vaccination history.

Seroprotected Participants	First Vaccine (91 Participants)			Repeated Vaccine (32 Participants)		
	H1N1	H3N2	B1	H1N1	H3N2	B1
**Day 0**	0 (0.0%)	0 (0.0%)	0 (0.0%)	0 (0.0%)	0 (0.0%)	0 (0.0%)
**Day 7**	42 (46.15%)	34 (37.36%)	38 (41.76%)	12 (37.5%)	13 (40.62%)	11 (34.38%)
**Day 90**	59 (64.84%)	61 (67.03%)	71 (78.02%)	9 (28.13%)	15 (46.88%)	20 (62.5%)
**Neither Day 7 nor Day 90** **(non-responders)**	28 (30.77%)	26 (28.57%)	16 (17.58%)	18 (56.25%)	14 (43.75%)	9 (28.16%)
**Only Day 7** **(short-term responders)**	4 (4.4%)	4 (4.4%)	4 (4.4%)	5 (15.62%)	3 (9.37%)	3 (9.37%)
**Only Day 90** **(late responders)**	21 (23.08%)	31 (34.07%)	37 (40.66%)	2 (6.25%)	5 (15.62%)	12 (37.5%)
**Day 7 and Day 90** **(ideal responders)**	38 (41.76%)	30 (32.97%)	34 (37.36%)	7 (21.88%)	10 (31.25%)	8 (25.0%)

## Data Availability

The datasets that support the findings of this study are managed by Kyoto University Graduate School and Faculty of Medicine and can be requested from the authors.

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
