# Peer review of "Transcriptomic Analysis Reveals Sixteen Potential Genes Associated with the Successful Differentiation of Antibody-Secreting Cells through the Utilization of Unfolded Protein Response Mechanisms in Robust Responders to the Influenza Vaccine"

_vaccines, 2024, doi:10.3390/vaccines12020136_

Round 1
Reviewer 1 Report
Comments and Suggestions for Authors
This work assessed the vaccine immunogenicity and transcriptomic profiling with blood samples, trying to figure out the signature genes correlating to the strength and longevity of specific immune responses. I am not sure about the rightness of the use of "cross-protection" in this study, the protections against either influenza strain virtually relies on the immunogenicity of each antigen rather than the cross-protectivity of either H1, H3 or B/HA in the seasonal influenza vaccine.
The aspects that impact the elicitation of the specific immune responses are a lot and complicated. The extremely general analysis of gene transcription using the principle component analysis method could show the obvious separation of some genes from the fully protected group. Some genes show the significantly differential expression, like MZB1 and XBP1, and seems as the gene signatures correlated to the full protection. Maybe, those outstanding gene activations could provide interesting targets for testing and evaluating the strength of immune responses in East Asian people. I think it would be difficult to draw a further rigorous conclusion about their correlations to better protection, otherwise there is a great liklihood this study can mislead the relevant research.
Minor suggestions are as follows.
- The title is quite long, and suggest to shorten it in not more than three lines.
- In line 84, "We classified participants into eight groups,..." not nine groups.
Reviewer 2 Report
Comments and Suggestions for Authors
Good morning for the all authors,
Analyzing the manuscript with ID: vaccines-2823596 - peer-review-v1, entitled "Transcriptomic analysis reveals 16 potential genes associated with the successful differentiation of antibody-secreting cells through the utilization of unfolded protein response mechanisms in robust responders to the influenza vaccine" which to be published in the Journal Vaccines,
I consider that:
1. The authors of the article proposed a much-discussed topic in today's medical scientific world, namely: the effectiveness of vaccines and especially influenza vaccines.
2. The article follows all the specific instructions of the journal presented in: aims and scope, instructions for authors and other information about the journal.
3. Introduction Chapter: the authors present relevant information for the chosen subject.
4. Materials and Methods Chapter: the data presented in this manuscript are very well structured and coherent.
5. Results Chapter: the methods, statistical analysis and results are well presented and easy to understand in the text of the manuscript. Tables 1-2, Figures 1-5 and Supplementary Table 1-5_V1 are suggestive of the study carried out and very well presented.
6. In the Discussions Chapter, the authors exemplified and compared all the results obtained by them with the studies and assessments of other authors, according to the bibliography.
7. The bibliography chosen by the authors corresponds to the requirements and refers to the subject of the article.
8. All authors have made a fair contribution to the study.
9. The authors also received funding for this study (grant 778 n°ANR-10-LABX-62-IBEID) demonstrating professionalism and medical scientific relevance.
In conclusion:
I ACCEPT for publication of the Journal Vaccines this article.
Congratulations on this article!
Author Response
Dear Reviewer,
We sincerely appreciate your dedicated effort in reviewing this manuscript and we would like to thank you very much for the valuable and constructive feedback that you have graciously shared with us.
Kind regards
Reviewer 3 Report
Comments and Suggestions for Authors
Ahmed Tawfik and colleagues have performed a descriptive study following trivalent Influenza vaccination of immunologically naive individuals. They categorise individuals into different groups dependent on their immune status following vaccination and perform bulk transcriptomic analysis pre-vaccination and at day seven. They describe the upregulated genes in individuals who had an optimal response (detectible immune responses to all strains included in the vaccine at day seven and day ninety). They compared the bulk genes upregulated in the complete responders to the other individuals who did not have an optimal response. They perform a short case study on each gene they identify in their analysis to shed light on the transcriptomic variation that could explain the differences in vaccine response. They define two pathways, one involving B cell proliferation and the other involving metabolic processes to allow for the development of ASCs.
I found the paper to be well-written. Though much of the data is highly descriptive, and not all of their findings are entirely novel, the authors cite previous studies that agree with their findings. As is the nature of this kind of study, the data presented was very descriptive, with some sections of the results reading somewhat like a review. However, I understand why the authors chose to do this to place their findings in the context of the broader literature. Overall, their findings seem robust and are of interest.
Specific comments
The non-responders you identify may be just late responders who lose their response by day 90, so I don’t believe these can be referred to as non-responders. Perhaps no detectable response would be better wording. Some extra time points within the study would have helped address this in finer detail.
I do not think that the use of the word protected throughout the manuscript is correct. While this is based upon the consensus that the HAI assay detects a protective response at a certain level (as stated in your methods), these individuals have not been challenged with Influenza, and we cannot determine that they are indeed protected. Perhaps changing the wording from protected to responders would be more correct (e.g. in Figure 1, the green group would be full responders rather than full protection).
More assays for the serological data would be useful. While in widespread use, HAI is not very sensitive and corroborating findings with ELISA or flow cytometry analysis would improve this data.
Lines 632-640 When discussing this section, you mention that the lack of an immune response from the vaccine is expected as they had a zero baseline HAI. However, immune B cell memory may have been laid down previously despite having no detectible HAI response at baseline. I would expect previously vaccinated individuals to respond quicker and more robustly to subsequent vaccination. Do you have any data on whether their previous vaccination was homologous to the old vaccine?
Did you try linking any of these findings to the metadata you describe in your methods? It would be interesting to know if any transcriptomic differences could be linked to age or sex (for example).
Reviewer 4 Report
Comments and Suggestions for Authors
Review of MS# 2823596 for Vaccines.
This paper performs a transcriptomic analysis following vaccination with tri-valent influenza vaccine in a Japanese population during the 2011/2012 flu season.
4 groups are identified: non-responders, short-term responders, late-responders, and idea responders.
The analysis indicates that changes in expression of 16 genes are specifically associated with an idea vaccine response. Namely one which gives immediate and durable response to all three antigens.
These can be grouped into classes of protein involved in B cell proliferation, unfolded protein response and cellular haemostasis.
The experiments are well described, use appropriate methodology and the conclusions drawn are supported by the data provided.
The paper is generally well written and easy to follow for the most part. However, the section detailing the characteristics of the 16 genes (pages 9 – 14) becomes extremely repetitive. It would be much clearer to formulate the numerical values in a table comparing the genes and responder groups.
The subject is of interest to the readership of Vaccines and in my opinion the paper should be suitable for publication, after a revision to make this simplification.
Comments on the Quality of English LanguageGenerally good. Pages 9 - 16 become very repetitive, if possible please simplify.
